# Impact of COVID-19 on hospital admission of acute stroke patients in Bangladesh

**A. T. M. Hasibul Hasan** [1]*, **Subir Chandra Das**[1], **Muhammad Sougatul Islam**[2], **Mohaimen Mansur**[3], **Md. Shajedur Rahman Shawon**[4], **Rashedul Hassan**[5], **Mohammad Shah Jahirul Hoque Chowdhury**[1], **Md. Badrul Alam Mondal**[1], **Quazi Deen Mohammad**[1]

**1** Department of Neurology, National Institute of Neurosciences & Hospital, Dhaka, Bangladesh, **2** BioTED, Dhaka, Bangladesh, **3** Institute of Statistical Research and Training, University of Dhaka, Dhaka, Bangladesh, **4** Center for Big Data Research in Health, UNSW Medicine, NSW, Australia, **5** Department of Medicine, Green Life Medical College & Hospital, Dhaka, Bangladesh

* parag007us@gmail.com

## Abstract

### Background

With the proposed pathophysiologic mechanism of neurologic injury by SARS CoV-2, the frequency of stroke and henceforth the related hospital admissions were expected to rise. This paper investigated this presumption by comparing the frequency of admissions of stroke cases in Bangladesh before and during the pandemic.

### Methods

This is a retrospective analysis of stroke admissions in a 100-bed stroke unit at the National Institute of Neurosciences and Hospital (NINS&H) which is considerably a large stroke unit. All the admitted cases from 1 January to 30 June 2020 were considered. Poisson regression models were used to determine whether statistically significant changes in admission rates can be found before and after 25 March since when there is a surge in COVID-19 infections.

### Results

A total of 1394 stroke patients took admission in the stroke unit during the study period. Half of the patients were older than 60 years, whereas only 2.6% were 30 years old or younger. The male to female ratio is 1.06:1. From January to March 2020, the mean rate of admission was 302.3 cases per month, which dropped to 162.3 cases per month from April to June, with an overall reduction of 46.3% in acute stroke admission per month. In those two periods, reductions in average admission per month for ischemic stroke (IST), intracerebral hemorrhage (ICH), subarachnoid hemorrhage (SAH) and venous stroke (VS) were 45.5%, 37.2%, 71.4% and 39.0%, respectively. Based on weekly data, results of Poisson regressions confirm that the average number of admissions per week dropped significantly during the last three months of the sample period. Further, in the first three months, a total of 22 cases of hyperacute stroke management were done, whereas, in the last three months, there was an 86.4% reduction in the number of hyperacute stroke patients getting

**Data Availability Statement:** All relevant data are within the paper and its Supporting Information files.

**Funding:** The author(s) received no specific funding for this work.

**Competing interests:** The authors have declared that no competing interests exist.

reperfusion treatment. Only 38 patients (2.7%) were later found to be RT-PCR SARS Cov-2 positive based on nasal swab testing.

## Conclusion

This study revealed a more than fifty percent reduction in acute stroke admission during the COVID-19 pandemic. Whether the reduction is related to the fear of getting infected by COVID-19 from hospitalization or the overall restriction on public movement or stay-home measures remains unknown.

## Introduction

Though COVID was initially reported as a case of atypical pneumonia from Wuhan, China in December 2019, it was subsequently found to involve other systems as well, especially the nervous system [1, 2]. Neurologic dysfunction is reported in up to one-third of the cases of COVID-19 patients [3]. The frequency of stroke has been reported to range from 2.8% to 5.7% among confirmed and hospitalized COVID-19 patients [3–5]. Reported COVID-19-related hemorrhagic strokes are far less common than associated ischemic strokes [6–9]. While the pathogenesis of COVID-19-related hemorrhagic strokes is still not fully known, hypercoagulable state, vasculitis and cardiomyopathy had been suspected as potential pathogenic mechanisms for ischemic stroke in COVID-19 patients [10, 11]. Some researchers further stated that the viral affinity to the ACE-2 receptor present in endothelium might be responsible for the rupture of intracranial vessel wall [12]. COVID-19-related stroke patients were more likely to be older, hypertensive, and had a higher D-dimer level [13]. Qin C et all [14] reported that COVID-19 related stroke patients had more comorbidity, lower platelet counts and leukocyte counts, and the patients had higher levels of D-dimers, cardiac troponin I, NT pro-brain natriuretic peptide, and interleukin-6.

Considering the pandemic nature of the spread of COVID-19, together with the proposed pathogenic mechanism of stroke in its patient, the number of acute stroke cases at hospitals was expected to rise sharply. But in recent months, neurologists from different parts of the world have reported a reasonable drop in the volume of acute stroke patients showing up at emergency care [15]. COVID-19 was first reported in Bangladesh on March 8, 2020 by the Institute of Epidemiology, Disease Control and Research (IEDCR). Soon after the virus was detected in the country, a rise in the infection rate was observed since early April. As of 15 June, the attack rate (AR) in Bangladesh is 532.1 per million [16]. For the last several weeks the case detection rate has been more than 20% and the total number of cases has exceeded two hundred and seventy-five thousand on 16 August 2020 [17]. With the spread of the outbreak, a large number of stroke cases were concurrently expected to get admitted to hospitals for treatment.

The current study was conducted at the stroke unit of the National Institute of Neuroscience & Hospital (NINS&H), Dhaka, Bangladesh. NINS&H is the center of excellence and the only dedicated center to provide comprehensive stroke care in Bangladesh. Its 100-bed stroke unit is one of the largest of this kind in the world.

This study assesses whether the volume of actual stroke admissions during the pandemic match the suspected rise in stroke cases by conducting a trend analysis of stroke admission at the NINS&H.

## Materials and methods

This is a retrospective analysis of all stroke patients admitted to the stroke unit of NINS&H from the 1st January to the 30th June 2020. All cases of stroke were confirmed by CT scans and/ or MRI imaging studies. Male and female patients aged 18 years or above who had confirmed diagnosis of stroke were admitted to the stroke unit. The hospital had not been declared as a Government designated COVID-19 hospital during the study period.

### The stroke unit and stroke assessment

The stroke unit in NINS&H has fifty beds for male patients and fifty beds for female patients. This is one of the largest comprehensive stroke care facilities in the world and provides all kinds of acute stroke interventions. Patients from all corners of the country seek treatment in this center and are often referred there for treatment by doctors. The criteria for admission in the stroke unit is a CT scan and/or an MRI confirming an incidence of stroke that requires attending the emergency within 48 hours of stroke onset. Any case of stroke mimic was excluded from the study.

### Assessment of COVID-19 status

All patients arriving at the emergency are examined for any symptom of COVID-19 via different tests including an evaluation of chest X-ray before admission. If any patient of stroke developed shortness of breath or showed an oxygen saturation level below 90%, RT-PCR for SARS CoV-2 from nasal swab was done. Patients with positive RT-PCR reports were referred to the Government-designated COVID-19 hospitals for treatment.

### Ethical issues

Prior to the commencement of the study, the protocol was reviewed and approved by the Ethical Review Committee (ERC), National Institute of Neurosciences and Hospital. All the data were collected manually from the hospital records and were fully anonymized. The need for informed consents from patients was also waived by the ERC due to the retrospective nature of the study. This study did not involve any intervention or experiment that may cause any harm to the study patients or any other human beings. The privacy and confidentiality of patient information were strictly maintained throughout the study.

### Data analysis

Both monthly and weekly admission rates were considered for data analysis. In the case of the week-wise distribution of data 12 weeks before 25 March 2020 were considered as the pre-COVID period and the 12 weeks onwards were considered as the COVID period. Following the chosen cut-off date of 25 March Bangladesh began to experience a substantial rise in confirmed COVID-19 cases which saw a jump from mere 39 cases on that date to nearly 5000 cases in four weeks [16]. Data analysis was done by using Statistical Package for Social Sciences (SPSS) version 21 and R version 4.0.0. In addition to monthly frequency distributions and time series plots, Poisson regressions are used to evaluate the data for significant changes in weekly admission rates before and during the COVID-19 period.

## Results

A total of 1394 stroke patients were admitted to the stroke unit of the National Institute of Neurosciences and Hospital during the study period. Among them, 38 patients (2.7%) were later found to be RT-PCR positive for SARS Cov-2 based on nasal swab testing. The percentage

**Table 1. Distribution of patients by age and sex (n = 1394).**

| Characteristics | Groups | Number (n) | Percentage (%) |
|---|---|---|---|
| Age (in years) | | | |
| | <30 | 37 | 2.65 |
| | 30–60 | 660 | 47.34 |
| | >60 | 697 | 50.00 |
| Sex | | | |
| | Male | 720 | 51.60 |
| | Female | 674 | 48.40 |

distribution presented in Table 1 shows that half of the patients were older than 60 years, whereas only 2.6% were 30 years old or younger. The male-to-female ratio was 1.06:1.

Fig 1 contains plots of monthly time series of the number of admissions for different types of stroke cases. A common pattern is that there is a large drop after March when the number of confirmed COVID-19 cases began to surge in Bangladesh. From January to March (the pre-COVID period), a total of 907 acute stroke cases were admitted to the stroke unit with a mean rate of admission of 302 cases per month, whereas from April to June (the COVID period) total admission was only 487 with a mean rate of admission of 162.3 cases per month. This implies an overall reduction of 46.3% in acute stroke admission per month. For each type of stroke, the number of admissions during the pre-COVID period is higher than that during the COVID period, as revealed by Fig 2. The mean admission rate for ischemic stroke (IST), intra-cerebral hemorrhage (ICH), subarachnoid hemorrhage (SAH) and venous stroke (VS) before and during the pandemic were 78.3 versus 42.7; 149.7 versus 94; 60.7 versus 17.3 and 13.7 versus 8.3 cases per month, respectively. The reductions in monthly mean admissions for IST, ICH, SAH and VS were 45.5%, 37.2%, 71.4% and 39.0%, respectively.

To investigate the problem further, weekly admission data were analyzed in addition to monthly data. Starting from the 1st of January 2020, admissions over 24 consecutive weeks were considered. The first 12 weeks of the study period were considered as the pre-COVID period and the last 12 weeks as the COVID period. For each stroke type, Poisson regression was run, where the dependent variable was the count of admissions of stroke patients and the independent variable was a COVID period indicator (0 = pre-COVID, 1 = COVID). The results of the regressions are reported in Table 2. Negative coefficients with p-values < 0.05

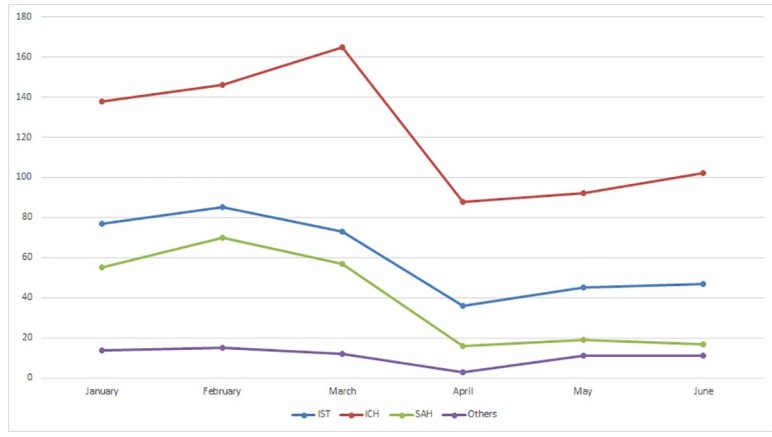

**Fig 1. Time trends in admission of various types of stroke patients (n = 1394).**

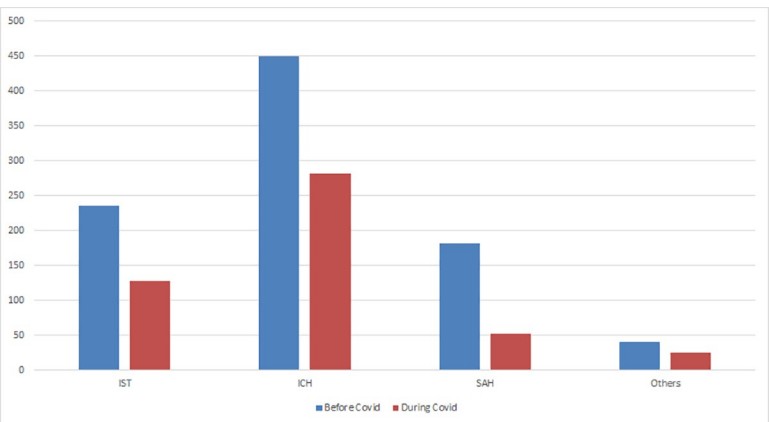

**Fig 2. Number of admitted stroke patients before and after Covid-19 (n = 1394).**

indicate statistically significant (at 5% level of significance) drops in the weekly rates of admission during the COVID period.

Robustness of regression results was assessed by further conducting a statistical test of the difference between two Poisson rates modified for small sample applications [18]. The null hypothesis of the test is that there is no difference in the weekly rate of stroke admissions during the pre-COVID period, $\lambda_1$, and the weekly rate during the COVIID period, $\lambda_2$. The alternative hypothesis is that $\lambda_2 < \lambda_1$. Results of the tests reported in Table 3 confirm rejection of the null hypothesis at least at a 5% level of significance and support the finding of the Poisson regression that the weekly rate of admission for all types of stroke patients reduced significantly during the COVID period compared to the pre-COVID period.

In the first three months, a total of 21 cases of hyperacute stroke were managed in the form of I/V thrombolysis in the stroke unit. The last three months, however, observed an 85.7% reduction in the number of hyperacute stroke patients receiving I/V thrombolysis and reperfusion treatment. This downward trend is shown in Fig 3.

**Table 2. Results of Poisson regressions for different stroke types.**

| Stroke type | | Estimate | Standard error | Z statistic | p-value |
|---|---|---|---|---|---|
| IST | | | | | |
| | Intercept | 2.546 | 0.081 | 31.486 | < 0.001 |
| | COVID | -0.277 | 0.123 | -2.249 | 0.025 |
| ICH | | | | | |
| | Intercept | 3.458 | 0.051 | 67.495 | < 0.001 |
| | COVID | -0.421 | 0.081 | -5.177 | < 0.001 |
| SAH | | | | | |
| | Intercept | 2.100 | 0.101 | 20.790 | < 0.001 |
| | COVID | -0.871 | 0.186 | -4.685 | < 0.001 |
| All | | | | | |
| | Intercept | 3.975 | 0.040 | 100.480 | < 0.001 |
| | COVID | -0.434 | 0.063 | -6.880 | < 0.001 |

*IST = Ischemic Stroke, ICH = Intracerebral Hemorrhage, SAH = Subarachnoid hemorrhage.

**Table 3. Results of a test for difference in the rates of weekly admission before and during the COVID period.**

| Stroke type | A test for the difference of two Poisson rates $H_0: \lambda_1 = \lambda_2$ vs $H_1: \lambda_2 < \lambda_1$ | |
|---|---|---|
| | Z statistic | p-value |
| IST | 2.26 | 0.012 |
| ICH | 5.24 | < 0.001 |
| SAH | 4.94 | < 0.001 |
| All | 6.97 | < 0.001 |

*IST = Ischemic Stroke, ICH = Intracerebral Hemorrhage, SAH = Subarachnoid hemorrhage.

## Discussion

This study assesses the impact of the COVID-19 outbreak on the admission of acute stroke patients in Bangladesh. As the pandemic hit the country, Bangladesh Government designated several hospitals of the country exclusively for the treatment of COVID-19 patients. These include the Dhaka Medical College Hospital which deals with a considerable portion of stroke cases each year. For this reason, the stroke unit of the National Institute of Neurosciences and Hospital, which was exempted from providing COVID-19 treatment, was expected to experience a higher-than-normal load of stroke patients during the spread of the pandemic.

While the exact pathophysiology of COVID-19 related stroke is still not fully known, there is growing evidence which links COVID-19 to coagulopathy causing venous and arterial thrombosis [10, 11]. There are also reports of several cases of large vessel occlusion in stroke patients with COVID-19 infections [11], and consequent increase in the incidence of related strokes during the peak of the pandemic [19]. A recent study further found a significant association between the severity of COVID-19 and increased risk of acute stroke [20]. Considering these findings, the number of admissions of acute stroke patients were reasonably expected to show a steep rise during the pandemic. In contrast to this speculation, our analysis, based on the number of admitted stroke cases in the largest stroke treatment center of the country over the period of January to June 2020, revealed a gross (46.3%) reduction in the rate of acute stroke admissions in last three months of the study period when the infection surged. The reduction for SAH admissions was the highest (71.4%) followed by IST (45.6%), VS (39.02%) and ICH (37.1%) admissions. Results of Poisson regressions confirm that reductions in weekly

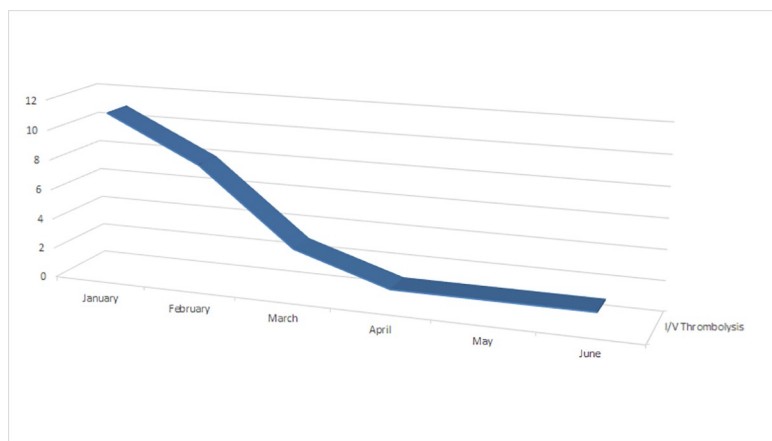

**Fig 3. Time trend in I/V thrombolysis (n = 1394).**

admission rates during the COVID period are statistically significant. Hoyer et al [21] reported a similar picture of acute stroke admission during the pandemic based on data from four academic stroke centers in Germany. The study compared stroke admission rates in the same weeks of the year 2019 and 2020 and reported 38–46% drops in acute stroke admissions in those centers. It assumed that while factors such as the size of stroke center's catchment area might be responsible for differential reductions in the four centers, the fear of getting infected at hospitals and forced lock-down measures might have resulted in lower interest in seeking medical help and a consequent overall reduction in stroke admissions [21]. In agreement with Dhand A et al [22], Hoyer et al [21] further opined that social distancing and confinement at home might have resulted in delayed disclosure of symptoms, and late or no hospital admission following a wait-and-watch strategy. Similar trends in admission were also observed for other diseases. For example, Stöhr E et al [23] reported a significant reduction in hospitalization for all cardiovascular events in Germany immediately after the Government imposed restriction on movement and social distancing measures.

COVID-19 infected stroke admissions in our study were only 2.7% of the total admission. This percentage figure is similar to the finding reported in Qin C et al [14] which is based on one hospital from Wuhan, China, but three times the figure reported in Yaghi S et al [24], a study based in New York. Bangladesh did not impose any nation-wide lock-down following the detection of its first COVID-19 case on the 8th of March, 2020. Only on March 26, a nation-wide public holiday was declared and public transport was placed under restriction. These measures were subsequently extended for several weeks. However, movements of ambulance or other emergency services were exempted from these restrictions. Therefore, it is unlikely that the reduction in admitted stroke cases is due to the restriction on movements or closure of common public transports. It is also not clear whether the fear of getting infected at hospitals or the overall 'stay home' publicity, in general, is the main cause of this reduction. While deaths and detrimental long-term effects including impaired speech, paralysis of limbs and restricted physical abilities were speculated for missing timely hospital treatment, the extent of such consequences could not be confirmed due to the unavailability of required data.

This study has a few limitations. First, this is a retrospective analysis based on hospital records. Therefore, further analyses beyond trend exercises, such as investigating reasons behind the reduction in stroke admissions, were not possible. Second, since the stroke unit at NINS&H started its journey only in September 2019, we could not compare stroke admissions in the same months of 2019 and 2020. This would have ensured a fairer comparison devoid of any possible seasonal effects on admission. The major strength of the study, however, is that a considerable number of admitted stroke cases were analyzed over a substantially long period covering both periods before and during the COVID-19 pandemic. Importantly, the data came from one of the largest stroke units in the world which is also the highest center of referral for stroke cases in Bangladesh providing 24X7 comprehensive stroke care.

## Conclusion

In contrary to the presumption that stroke burden may increase in hospitals during the COVID-19 pandemic, our study found a more than fifty percent reduction in the number of acute stroke admissions in the largest stroke unit in Bangladesh. The decline was the most prominent in SAH admissions followed by IST admissions. Thus, this study is another addition to the recent evidence of such decreasing trends in stroke admissions during the pandemic. Whether the reduction is related to the fear of getting infected by COVID-19 from hospitalization or the overall restriction on public movement or stay-home measures needs to be investigated by further studies. Even in this time of the pandemic and consequent social

distancing measures, it is crucial for patients experiencing acute stroke attacks to seek immediate medical help so that timely diagnosis, intervention and management of stroke are possible. Increasing public awareness in this regard will help avoid unnecessary and unfortunate losses of lives from untreated strokes.

## Supporting information

**S1 Data.**
(DOCX)

## Acknowledgments

We acknowledge the contribution of our supporting staffs at the stroke unit, NINS&H for helping us with the necessary data for the study.

## Author Contributions

**Conceptualization:** A. T. M. Hasibul Hasan, Muhammad Sougatul Islam, Rashedul Hassan.

**Data curation:** A. T. M. Hasibul Hasan, Subir Chandra Das, Mohaimen Mansur, Mohammad Shah Jahirul Hoque Chowdhury.

**Formal analysis:** Muhammad Sougatul Islam, Mohaimen Mansur, Md. Shajedur Rahman Shawon.

**Investigation:** Subir Chandra Das.

**Methodology:** A. T. M. Hasibul Hasan, Muhammad Sougatul Islam, Mohaimen Mansur.

**Project administration:** Subir Chandra Das, Mohaimen Mansur, Rashedul Hassan, Mohammad Shah Jahirul Hoque Chowdhury, Md. Badrul Alam Mondal.

**Resources:** Subir Chandra Das, Muhammad Sougatul Islam, Mohaimen Mansur, Rashedul Hassan, Md. Badrul Alam Mondal.

**Supervision:** A. T. M. Hasibul Hasan, Muhammad Sougatul Islam, Mohammad Shah Jahirul Hoque Chowdhury, Md. Badrul Alam Mondal, Quazi Deen Mohammad.

**Writing – original draft:** A. T. M. Hasibul Hasan.

**Writing – review & editing:** Muhammad Sougatul Islam, Mohaimen Mansur, Md. Shajedur Rahman Shawon.

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
