## [Decision Letter · Decision Letter 0]

20 Oct 2020

PONE-D-20-29228

Impact of COVID-19 on Hospital Admission of Acute Stroke patients in Bangladesh

PLOS ONE

Dear Dr. Hasan,

Thank you for submitting your manuscript to PLOS ONE. After careful consideration, we feel that it has merit but does not fully meet PLOS ONE’s publication criteria as it currently stands. Therefore, we invite you to submit a revised version of the manuscript that addresses the points raised during the review process.

We look forward to receiving your revised manuscript.

Kind regards,

Wen-Jun Tu

Academic Editor

PLOS ONE

Journal Requirements:

2.Thank you for stating the following in the Financial Disclosure section:

[The author(s) received no specific funding for this work.].   

We note that one or more of the authors are employed by a commercial company: BioTED

Reviewers' comments:

Reviewer's Responses to Questions

**Comments to the Author**

1. Is the manuscript technically sound, and do the data support the conclusions?

Reviewer #1: Yes

Reviewer #2: Yes

Reviewer #3: No

2. Has the statistical analysis been performed appropriately and rigorously? 

Reviewer #1: Yes

Reviewer #2: Yes

Reviewer #3: No

3. Have the authors made all data underlying the findings in their manuscript fully available?

Reviewer #1: Yes

Reviewer #2: Yes

Reviewer #3: Yes

4. Is the manuscript presented in an intelligible fashion and written in standard English?

Reviewer #1: Yes

Reviewer #2: No

Reviewer #3: Yes

5. Review Comments to the Author

Reviewer #1: 1. Need to have the manuscript reviewed by a science/technical writer would be useful as the discussion section has a lot of verbiage.

2. There are typographical errors which can be addressed by spell check

Reviewer #2: The manuscript requires some grammatical assistance.

The overall information regarding stroke admissions pre and during COVID is of value and interest to the scientific and medical community. Although limited application as only one country- still important.

Would like the authors to comment whether these individuals who didn't come to hospital- what do the authors anticipate the long term effects of people not coming to the hospital with stroke? Was there a rise in stroke related deaths? Do the authors know this?

Reviewer #3: While the concept is interesting, the epidemiology methods are not rigorous and the study has not been performed in a manner that will actually answer the question. Currently, total infection rates are not known, comparisons across different time periods may not be valid as SARS-CoV-2 could have been present well before March, and the pathophysiology of COVID-19 related stroke is not well known yet, so it's not clear how long after an infection that a stroke may develop.

6. PLOS authors have the option to publish the peer review history of their article (what does this mean?). If published, this will include your full peer review and any attached files.

Reviewer #1: No

Reviewer #2: No

Reviewer #3: No

---

## [Author Response · Author response to Decision Letter 0]

29 Nov 2020

Response to Reviewers

Reviewer #1: 

Comment 1: Need to have the manuscript reviewed by a science/technical writer would be useful as the discussion section has a lot of verbiage.

Authors’ response: We thank the reviewer for his/her advice. We have now rewritten the discussion section to remove the verbiage as much as possible and have the manuscript reviewed by a professional technical writer.

Comment 2: There are typographical errors which can be addressed by spell check

Authors’ response: We apologize for overlooking the errors. We have now corrected the typographical errors we have identified.

Reviewer #2: 

Comments: The manuscript requires some grammatical assistance.

The overall information regarding stroke admissions pre and during COVID is of value and interest to the scientific and medical community. Although limited application as only one country- still important. Would like the authors to comment whether these individuals who didn't come to hospital- what do the authors anticipate the long-term effects of people not coming to the hospital with stroke? Was there a rise in stroke related deaths? Do the authors know this?

Authors’ response: We thank the reviewer for this important suggestion. We have identified and corrected some grammatical mistakes after careful revision of the manuscript, and further had it reviewed by a professional proof reader. 

Like the reviewer we were also curious about the fate of the individuals who possibly failed to turn up into hospitals for stroke treatment during the pandemic. To our knowledge, no records of these individual were available nor any information on them including mortality were collected. Unfortunately, therefore, we are not in a position to confirm any definitive consequence of missing timely hospital treatment. However, we speculate fatalities or detrimental long-term effects including impaired speech and paralysis for these individuals. 

We have now included the following statement in the Discussion section:

“While deaths and detrimental long-term effects including impaired speech, paralysis of limbs and restricted physical abilities were speculated for missing timely hospital treatment, the extent of such consequences could not be confirmed due to unavailability of required data.”

Reviewer #3: 

Comments: While the concept is interesting, the epidemiology methods are not rigorous and the study has not been performed in a manner that will actually answer the question. Currently, total infection rates are not known, comparisons across different time periods may not be valid as SARS-CoV-2 could have been present well before March, and the pathophysiology of COVID-19 related stroke is not well known yet, so it's not clear how long after an infection that a stroke may develop.

Authors’ response: We thank the reviewer for the careful observation and comment.

In response, we first we want to make it clear that the only question investigated in this paper is whether the stroke admission data supported the general fear that there was a rise in stroke cases during the time (after March 2020), which saw a surge in COVID-19 infections in Bangladesh. The fact that an incidence of stroke requires special medical treatment available only at stroke caring facilities/hospitals it is reasonable to expect that admission data will fairly reflect major changes in the number of new stroke cases. It is evident from the data and analysis that, contrary to the fear, there is a significant drop in stroke admissions during the wide-spread outbreak of the virus compared to the earlier period.

We understand the reviewer’s concern regarding the validity of the study design for answering the research question. While we know that a prospective study following hospital admitted COVID-19 patients up to the development of any stroke would be a much better design, retrospective studies have been used to investigate similar questions (e.g., Hoyer et al., 2020). We, however, cited comparing just six months of the same year 2020 as a possible limitation of our study. We also acknowledged that comparison across the same periods of the COVID-19-year of 2020 and an earlier year, as conducted in Hoyer et al. (2020), would have been a fairer comparison. Nonetheless, we could not pursue this due to the unavailability of the required longer time series from a relatively young stroke unit and clearly pointed out this limitation in the manuscript. 

We want to further argue that the retrospective comparison of admissions before and including March (our pre-COVID period) and the period thereafter (our COVID period) is a reasonable, if not ideal, comparison. First, while the reviewer is correct in pointing out the possibility of the presence of SARS-COV-2 in Bangladesh before March, we have strong reasons to believe that the total number of infections prior to March was much lower than that after March since when official statistics reported a rapid increase in the number of infections in the country. 

Second, while it is true that the pathophysiology of COVID-19 related stroke is still not perfectly known, we found a fair amount of evidence (as cited in the paper) which links COVID-19 to coagulopathy causing venous and arterial thrombosis (Oxley et al., 2020, Beyrouti et al., 2020) which are major risk factors for stroke. A recent study further found significant association between the severity of COVID-19 and increased risk of acute stroke (Siepman et al., 2020). 

Most importantly, although the exact duration of development of stroke in COVID-19 patients may not be definitely known, the study by Beyrouti et al. (2020) found that five out of six patients admitted with both COVID-19 and stroke actually developed ischaemic stroke within 8-24 days after COVID-19 symptom onset. Since we compared 3-month (90-day) average admissions before and after March we reasonably expected a significant number of new developments of stroke to occur during the latter period of higher rates of infection and get reflected accordingly in the admission data during that period. This made us believe that the comparisons we made are fair for testing our hypothesis. The fact that the empirical evidence goes in the opposite direction of the hypothesis only made it even more interesting.

We have already cited the references used in making our arguments in the original manuscript and included further evidence.

References:

1. Hoyer, C., Ebert, A., Huttner, H.B., Puetz, V., Kallmünzer, B., Barlinn, K., Haverkamp, C., Harloff, A., Brich, J., Platten, M. and Szabo, K., 2020. Acute Stroke in Times of the COVID-19 Pandemic: A Multicenter Study. Stroke, pp.STROKEAHA-120.

2. Oxley, T.J., Mocco, J., Majidi, S., Kellner, C.P., Shoirah, H., Singh, I.P., De Leacy, R.A., Shigematsu, T., Ladner, T.R., Yaeger, K.A. and Skliut, M., 2020. Large-vessel stroke as a presenting feature of Covid-19 in the young. New England Journal of Medicine, 382(20), p.e60.

3. Beyrouti, R., Adams, M.E., Benjamin, L., Cohen, H., Farmer, S.F., Goh, Y.Y., Humphries, F., Jäger, H.R., Losseff, N.A., Perry, R.J. and Shah, S., 2020. Characteristics of ischaemic stroke associated with COVID-19. Journal of Neurology, Neurosurgery & Psychiatry.

4. Siepmann, T., Sedghi, A., Simon, E., Winzer, S., Barlinn, J., de With, K., Mirow, L., Wolz, M., Gruenewald, T., Schroettner, P. and von Bonin, S., 2020. Increased risk of acute stroke among patients with severe COVID‐19: a multicenter study and meta‐analysis. European Journal of Neurology.

---

## [Decision Letter · Decision Letter 1]

15 Dec 2020

PONE-D-20-29228R1

Impact of COVID-19 on hospital admission of acute stroke patients in Bangladesh

PLOS ONE

Dear Dr. Hasan,

Thank you for submitting your manuscript to PLOS ONE. After careful consideration, we feel that it has merit but does not fully meet PLOS ONE’s publication criteria as it currently stands. Therefore, we invite you to submit a revised version of the manuscript that addresses the points raised during the review process.

ACADEMIC EDITOR:

1. In order to provide a more complete information to our readers on the topic, we would like to emphasize the importance to cross referencing very recent material on the same topic published in "PLoS ONE ".

2. Added “Cao, J., Tu, W. J., Cheng, W., Yu, L., Liu, Y. K., Hu, X., & Liu, Q. (2020). Clinical features and short-term outcomes of 102 patients with corona virus disease 2019 in Wuhan, China. *Clinical Infectious Diseases*.”in revision text. 

We look forward to receiving your revised manuscript.

Kind regards,

Wen-Jun Tu

Academic Editor

PLOS ONE

Reviewers' comments:

Reviewer's Responses to Questions

**Comments to the Author**

1. If the authors have adequately addressed your comments raised in a previous round of review and you feel that this manuscript is now acceptable for publication, you may indicate that here to bypass the “Comments to the Author” section, enter your conflict of interest statement in the “Confidential to Editor” section, and submit your "Accept" recommendation.

Reviewer #1: All comments have been addressed

Reviewer #2: All comments have been addressed

Reviewer #3: (No Response)

2. Is the manuscript technically sound, and do the data support the conclusions?

Reviewer #1: Yes

Reviewer #2: (No Response)

Reviewer #3: (No Response)

3. Has the statistical analysis been performed appropriately and rigorously? 

Reviewer #1: Yes

Reviewer #2: Yes

Reviewer #3: (No Response)

4. Have the authors made all data underlying the findings in their manuscript fully available?

Reviewer #1: Yes

Reviewer #2: Yes

Reviewer #3: (No Response)

5. Is the manuscript presented in an intelligible fashion and written in standard English?

Reviewer #1: Yes

Reviewer #2: Yes

Reviewer #3: (No Response)

6. Review Comments to the Author

Reviewer #1: They have tried to improve the submitted manuscript grammatically, in my humble opinion it could have improved further.

Reviewer #2: No further comments for the authors of this manuscript. They addressed all of my concerns in the revised document.

Reviewer #3: Thank you for your attention to all the reviewers's comments and questions. I believe that the authors have addressed all comments to their abilities.

7. PLOS authors have the option to publish the peer review history of their article (what does this mean?). If published, this will include your full peer review and any attached files.

Reviewer #1: No

Reviewer #2: No

Reviewer #3: No

---

## [Author Response · Author response to Decision Letter 1]

21 Dec 2020

Response to the Academic Editor: 

Comment 1: In order to provide a more complete information to our readers on the topic, we would like to emphasize the importance to cross referencing very recent material on the same topic published in "PLoS ONE ".

Authors’ response: We would like to thank the academic editor for the comment. We have added relevant points in the “Introduction and Discussion” section from the following three related articles published very recently in PLos ONE. The articles are cited in text and listed in the reference according to the following serial numbers: 

2. Campiglio L, Priori A. Neurological symptom in acute COVID-19 infected patients: A survey among Italian physicians. PLoS One. 2020 Sep; 15 (9): e0238159.

5. Jillella DV, Janocko NJ, Nahab F, Benameur K, Greene JG, Write WL, et al. Ischemic stroke in COVID 19: An urgent need for identification and management. PLoS One. 2020 Sep; 15 (9): e0239443.

23. Stöhr E, Aksoy A, Campbell M, Al Zaidi M, Öztürk C, Vorloeper J, et al. Hospital admissions during COVID-19 lock-down in Germany: Differences in discretionary and unavoidable cardiovascular events. PLoS One. 2020; 15 (11): e0242653. 

Comment 2: Added “Cao, J., Tu, W. J., Cheng, W., Yu, L., Liu, Y. K., Hu, X., & Liu, Q. (2020). Clinical features and short-term outcomes of 102 patients with corona virus disease 2019 in Wuhan, China. Clinical Infectious Diseases.” in revision text. 

Authors’ response: We would like to thank the academic editor for suggesting this important and timely article on clinical presentation and outcome of COVID patients. We have added a few more words in the introduction and cited the article as a reference in the revised text. 

Comment 3: Resubmit figures.

Authors’ response: We have uploaded the files to PACE digital diagnostic tool to covert to appropriate tif format and made the adjustments as per the requirement of the journal. 

Response to the Reviewers’ comments: 

Reviewer 1:

Comment: They have tried to improve the submitted manuscript grammatically, in my humble opinion it could have improved further.

Authors’ response: We would like to thank the reviewer for the comment. We can ensure that we have tried to improve the grammar of the manuscript to the best of our abilities. We are hopeful that it now meets the required standard of the journal. 

Reviewer 2:

Comment: No further comments for the authors of this manuscript. They addressed all of my concerns in the revised document.

Authors’ response: We thank the reviewer for his/her important comments raised previously and we are happy to learn that we managed to address all his/her concerns.

Reviewer 3:

Comment: Thank you for your attention to all the reviewers' comments and questions. I believe that the authors have addressed all comments to their abilities.

Authors’ response: We thank all the reviewers for their comments on the original manuscript. We would like to ensure that we tried to address each comment to the best of our abilities. We believe that the comments helped us greatly to improve the manuscript.

---

## [Editor Report · Decision Letter 2]

26 Dec 2020

Impact of COVID-19 on hospital admission of acute stroke patients in Bangladesh

PONE-D-20-29228R2

Dear Dr. Hasan,

We’re pleased to inform you that your manuscript has been judged scientifically suitable for publication and will be formally accepted for publication once it meets all outstanding technical requirements.

Kind regards,

Wen-Jun Tu

Academic Editor

PLOS ONE
---

## [Editor Report · Acceptance letter]

5 Jan 2021

PONE-D-20-29228R2 

Impact of COVID-19 on hospital admission of acute stroke patients in Bangladesh 

Dear Dr. Hasan:

I'm pleased to inform you that your manuscript has been deemed suitable for publication in PLOS ONE. Congratulations! Your manuscript is now with our production department. 

Kind regards, 

on behalf of

Dr. Wen-Jun Tu 

Academic Editor

PLOS ONE